# Exploring the Therapeutic Potential of Estrogen-Related Receptor γ Inverse Agonists in Atopic Dermatitis-like Lesions

**DOI:** 10.3390/ijms26146959

**Published:** 2025-07-20

**Authors:** Ju Hyeon Bae, Sijoon Lee, Jae-Eon Lee, Sang Kyoon Kim, Jae-Han Jeon, Yong Hyun Jeon

**Affiliations:** 1Preclinical Research Center, Daegu-Gyeongbuk Medical Innovation Foundation (K-MEDIhub), 80 Cheombok-ro Dong-gu, Daegu 41061, Republic of Korea; wngus7@kmedihub.re.kr (J.H.B.); sjlee1013@kmedihub.re.kr (S.L.); koof12@kmedihub.re.kr (J.-E.L.); ksk1420@kmedihub.re.kr (S.K.K.); 2Department of Internal Medicine, School of Medicine, Kyungpook National University Chilgok Hospital, Kyungpook National University, 807 Hoguk-ro, Buk-gu, Daegu 41404, Republic of Korea; ggoloo@hanmail.net

**Keywords:** atopic dermatitis, estrogen-related receptor gamma (ERRγ), ERRγ inverse agonist, DN200434

## Abstract

Estrogen-related receptor γ (ERRγ) has been reported to regulate various inflammation-related diseases. Herein, we attempted to evaluate the effects of DN200434 as a modulator for ERRγ in mice with atopic dermatitis (AD). Levels of mRNA and protein expression for ERRγ were evaluated in normal and DNCB-induced AD-diagnosed skin. The effects of DN200434 on the chemokines, inflammatory cytokines, and AKT/MAPK/NFκB pathway signaling were investigated in TNF-α/IFN-γ-treated HaCaT cells. DNCB-induced AD mice received DN200434 intraperitoneally for 10 days. Epidermal thickness at the dorsal aspect of the inflamed skin, spleen index, serum IgE levels, and proinflammatory cytokine levels in the skin lesions were measured. Histopathological evaluations, including assessments of epidermal hyperplasia, dermal inflammation, hyperkeratosis, folliculitis, and mast cell counts, were performed to confirm diagnostic features. Significant elevations in ERRγ expression at the RNA and protein levels were observed in DNCB-induced AD lesions. DN200434 suppressed chemokine and inflammatory cytokine expression and inhibited the elevated phosphorylation levels of AKT, ERK, p38, and NFκB in TNF-α/IFN-γ-treated HaCaT cells. Treatment with DN200434 alleviated DNCB-induced AD symptoms. The histopathological score and levels of infiltrated mast cells were also markedly lower in DN200434-treated AD mice than in vehicle-treated AD mice. Consistently, DN200434 reduced the serum IgE level and mRNA expression of TNFα and IL-6 in AD-diagnosed lesions. Collectively, our findings indicated the feasibility of ERRγ as a therapeutic target for the regulation of AD and that DN200434 can be a useful therapeutic agent in treating AD.

## 1. Introduction

Atopic dermatitis (AD) is a chronic inflammatory skin diseases induced by abnormal immune responses, resulting in intense itching and a compromised skin barrier [1]. Globally, approximately 15–20% children and 1–3% adults grapple with this ailment, and its incidence is increasing, particularly in economically disadvantaged regions [2]. Extensive research has enhanced our comprehension of the immune and molecular mechanisms driving AD, facilitating the discovery of promising therapeutic strategies. With the discovery of novel biomarkers for AD treatment, numerous therapeutic agents have been identified. However, the therapeutic outcomes remain unsatisfactory in both preclinical and clinical settings. Therefore, additional targets and novel agents are urgently needed to effectively treat AD.

The estrogen-related receptors (ERRα, ERRβ, and ERRγ) function as intrinsically active nuclear receptors and show a high degree of sequence homology with estrogen receptors [3]. ERR isoforms exhibit predominant expression in multiple organs, such as the heart, kidney, pancreas, and skeletal muscle [4]. Among ERRs, ERRγ contributes to mitochondrial function by optimizing bioenergetics, facilitating oxidative phosphorylation, and ensuring cellular energy homeostasis, which is particularly relevant in metabolic diseases and cancer [5,6,7,8,9,10]. Based on the findings of the importance of ERRγ in the onset and progression of various diseases, several researchers have discovered novel modulators with functional activity for the treatment of ERRγ-associated diseases [11,12,13]. Recently, we reported on the discovery of a novel modulator for ERRγ (DN200434) that is an orally active small molecule proven to be effective in treating atherosclerosis, pancreatitis, anaplastic thyroid cancer, and sorafenib-resistant hepatocellular carcinoma [14,15,16,17].

Given the promising therapeutic effects of our ERRγ modulator against several diseases, we hypothesized that the regulation of ERRγ using DN200434 is a reasonable way of controlling AD-related diseases. In this study, we explored the feasibility of DN200434 as a therapeutic agent for AD in vitro and in vivo.

## 2. Results

### 2.1. Increased Endogenous Expression of ERRγ in DNCB-Induced AD Skin Lesion

To evaluate the status of endogenous ERRγ expression in DNCB-induced AD lesions, skin lesions exposed to either vehicle or DNCB solution were collected, and quantitative real-time PCR and western blotting examination were conducted (Figure 1A). As shown in Figure 1B, the expression of ERRγ mRNA was markedly elevated in skin lesions treated with DNCB relative to those treated with the vehicle. Consistently, we can observe higher expression of ERRγ protein in DNCB-treated skin lesions compared with that in vehicle-treated skin lesions (Figure 1C,D). Based on these results, ERRγ is considered a suitable target for therapeutic strategies against DNCB-induced AD.

### 2.2. Reduction in the mRNA Level of Chemokines and Inflammatory Cytokines in TNF-α/IFN-γ-Treated HaCaT Cells by DN200434

TNF-α/IFN-γ-induced activation of HaCaT cells is commonly employed to identify therapeutic agents against AD. We first evaluated the effect of DN200434 on HaCaT cell viability at various concentrations using CCK-8 assay. As shown in Figure 2A, the results of CCK8 assay revealed less cytotoxicity up to 10 μM in both non-stimulated HaCaT cells and TNF-α/IFN-γ stimulated HaCaT cells, with no significant difference in cell viability between the two groups.

Subsequently, we examined the effects of DN200434 on chemokines and inflammatory cytokine expression such as RANTES, MDC, TARC, IL-6, and IL-8 in TNF-α/IFN-γ treated HaCaT cells. Treatment of TNF-α/IFN-γ drastically up-regulated the mRNA expression of RANTES, MDC, TARC, IL-8, and IL-6 in HaCaT cells compared to non-treated HaCaT cells (Figure 2B–F). However, DN200434 markedly decreased the expression levels of chemokines and inflammatory cytokines in TNF-α/IFN-γ-treated HaCaT cells dose-dependently.

### 2.3. DN200434 Inhibits the Upregulation of Phosphorylated AKT, MAPK, and NF-κB in TNF-α/IFN-γ-Treated HaCaT Cells

AKT/MAPK/NFκB signaling pathway has been found to be closely associated with the pathogenesis of AD and mass production of inflammation mediators in AD lesions. Therefore, we explored the effects of DN200434 on AKT, MAPK, and NFκB signaling pathway in TNF-α/IFN-γ-treated HaCaT cells. As seen in Figure 3A,B, treatment with TNF-α/IFN-γ increased the phosphorylation of AKT, ERK, and p38 in HaCaT cells. However, pre-treatment with DN200434 reduced the elevated levels of p-AKT, p-ERK, and p-p38 to basal levels. Similarly, TNF-α/IFN-γ stimulation upregulated p-NFκB expression in HaCaT cells, but this increase was significantly inhibited by pre-treatment with DN200434.

### 2.4. DN200434 Alleviated DNCB-Induced AD Symptoms

We explored the effects of DN200434 on mice with DNCB-induced AD, as illustrated in Figure 4A. Seven days after sensitization with 1.0% DNCB solution, the dorsal skin of the mice was re-sensitized with 0.5% DNCB solution, followed by oral administration of either DN200434 or prednisone as positive control. Application of DNCB to the skin induced AD-like lesions characterized by erythema, excoriation, and xerosis (Figure 4B). Interestingly, treatment with DN200434 significantly reduced dorsal thickness and AD-related symptoms, including dryness, scaling, erosion, and hemorrhage, in mouse skin (Figure 4C–E). The spleen index was lower in DN200434-treated AD mice than in vehicle-treated AD mice (Figure 4F). Similar to the effects observed with DN200434, prednisone restored skin inflammation, reduced the dermatitis score, and decreased the spleen index compared to vehicle-treated AD mice. The overall therapeutic effects of DN200434 were comparable to those of prednisone.

### 2.5. DN200434 Decrease mRNA Level of Proinflammatory Cytokines in AD Lesions and Levels of Serum IgE in Mice with DNCB-Induced AD

We subsequently explored the changes in the expression of proinflammatory cytokines in AD lesions and serum IgE levels in DNCB-induced AD mice following DN200434 treatment. As illustrated in Figure 5A,B, DN200434 significantly reduced the expression of IL-6 and TNFα in AD skin lesions. The reduction in proinflammatory cytokine levels induced by DN200434 was comparable to that observed with prednisone. Serum IgE levels were consistently lower in DN200434-treated AD mice than in vehicle-treated AD mice (Figure 5C). Prednisone exhibited a more pronounced inhibitory effect on IgE levels compared to DN200434 in vivo.

### 2.6. Reduction of Skin Hyperplasia and Mast Cell Infiltration in AD Lesions by DN200434

We examined the histopathological alternations and mast cell infiltration levels in the AD lesions of DN200434-treated AD mice using H&E and toluidine blue staining (Figure 6A,B). Histological analysis using H&E staining revealed the inflammatory cell infiltration, hyperplasia of the epidermis, and inflammation of the dermis, hyperkeratosis, and folliculitis in the skin in DN200434-treated AD mice compared with control mice. However, treatment with DN200434 improved DNCB-induced histological alterations and significantly decreased dorsal skin thickness compared to that in the AD group (Figure 6C,D). Toluidine blue staining also demonstrated that DN200434 effectively reduced the number of infiltrated mast cells in the inflamed lesions of AD mice (Figure 6E). Although the difference was not statistically significant, the histological score and mast cell infiltration level were lower in DN200434-treated mice than in prednisone-treated mice. The epidermal thickness in DN200434-treated mice was comparable to that observed in prednisone-treated mice.

## 3. Discussion

ERRγ regulates important physiological functions in various metabolic diseases, and its pathophysiological relationship has been demonstrated in several reports [4]. However, the role of ERRγ in the regulation of AD has not been investigated to date. Here, we demonstrated that ERRγ expression is significantly upregulated in DNCB-induced AD lesions, and that inhibition of ERRγ by its modulator DN200434 leads to anti-inflammatory response in TNF-α/IFN-γ-treated HaCaT cells and in vivo DNCB-induced AD mice model via regulation of the AKT/MAPK/NF-κB signaling pathway.

The epidermis, the outermost layer of the skin, consists of keratinocytes at different stages of differentiation and serves as a protective barrier against external factors, including antigens and pathogens [18]. Elevated levels of chemokines (RANTES, TARC, and MDC) and inflammatory cytokines have been observed in the skin lesion of patients with AD. In this study, HaCaT cells were activated with IFN-γ and TNF-α, a widely used method for inducing inflammation in in vitro skin experiments. As expected, stimulation of human keratinocyte HaCaT cells with IFN-γ/TNF-α led to a significant increase in mRNA levels of chemokines such as RANTES, MDC, and TARC as well as inflammatory cytokines, including IL-8 and IL-6. However, pretreatment with DN200434 significantly downregulated their mRNA expression levels. These findings indicate that DN200434 has an ability with respect to immunomodulatory activity against AD lesions.

Numerous studies have demonstrated that the AKT/MAPK signaling pathways are closely related to the development of AD pathogenesis and expression of various pro-inflammatory genes [19]. Moreover, both AKT and MAPK regulate the NF-κB signaling pathway, which is critical for initiating various inflammatory diseases. Therefore, we sought to determine whether DN200434 could reduce AD-like skin inflammation in HaCaT cells by modulating the AKT/MAPK/NF-κB signaling pathway. We found that TNF-α/IFN-γ noticeably increased the levels of phosphorylated AKT, p38, and ERK in HaCaT cells. However, DN200434 treatment reduced these elevated phosphorylation levels. Furthermore, DN200434 effectively downregulated the levels of phosphorylated NF-κB, a key mediator of inflammatory activation, in TNF-α/IFN-γ-treated HaCaT cells. These findings suggest that the anti-AD effects of DN200434 are induced via the modulation of the AKT/MAPK/NF-κB signaling pathway.

For the clinical application of DN200434 in AD, establishing a relationship between therapeutic biomarker expression levels and AD is crucial. Interestingly, we identified an inverse correlation in ERRγ expression between normal skin lesions and DNCB-induced AD skin lesions, suggesting its relevance as a therapeutic target for AD.

Based on the in vitro and in vivo findings, we attempted to investigate whether DN200434 exhibited a therapeutic effect against AD in vivo. DNCB, a well-known allergen, is frequently employed to trigger AD in experimental models [20]. Furthermore, prednisone was used as a positive control for atopic dermatitis (AD) treatment when evaluating the effects of DN200434 in an in vivo AD model. Consistent with other reports, repeated exposure to DNCB led to hyperkeratosis, or a thick epidermis, and recruitment of lymphocytes and mast cells. However, DN200434 treatment drastically attenuated the DNCB-induced AD-symptoms and led to improvement in skin lesion severity and dorsal and epidermal layer thickness in DNCB-exposed mice. The spleen is a major organ that serves as the center for both cellular and humoral immune responses in living organisms, and prolonged inflammation of the skin causes spleen enlargement [21]. In our case, we observed splenic enlargement in DNCB-induced AD; however, DN200434 treatment decreased this occurrence. Interestingly, the alleviation of AD symptoms in DN200434-treated mice was similar to that observed in prednisone-treated mice. These findings suggest that DN20043-mediated ERRγ modulation induced potent anti-AD activity in vivo.

Aggressive recruitment of inflammatory cells including eosinophils, T cells, basophils, and mast cells occurs simultaneously because AD is a skin disorder with a severe inflammatory reaction [22]. In our study, H&E staining revealed marked inflammatory infiltration in DNCB-induced AD lesions, which was significantly attenuated by DN200434. Consistent with these findings, we observed an increase in the number of infiltrating mast cells as a hallmark of AD, as determined by toluidine blue staining. In contrast, treatment with DN200434 significantly reduced mast cell infiltration. The inhibitory effect on mast cell infiltration induced by DN200434 was comparable to that of prednisone. These findings indicate that the anti-AD effects of DN200434 are likely related to effective inhibition of the influx of inflammatory cells into AD skin lesions.

The progression of AD is associated with the aggressive production of proinflammatory cytokines in inflamed lesions, as well as elevated IgE production in the serum, a common observation in patients with AD [23]. In our mouse models of AD, we also observed a significant up-regulation of IL-6 and TNFα expression at the mRNA level, along with an increase in serum IgE levels. However, the administration of DN200434 suppressed the upregulated expression of IL-6 and TNFα in AD inflamed lesions and reduced serum IgE levels. The levels of IgE and proinflammatory cytokine inhibition were comparable between DN200434- and prednisone-treated mice. These findings suggest that the modulation of DN200434-mediated ERRγ induced potent anti-AD activity in vivo.

In conclusion, our findings showed that the modulation of ERRγ is a feasible approach for treatment of DNCB-induced AD, and DN200434, as its modulator, may be a promising candidate against AD in the future. Further investigations are warranted to explore the mode of action engaged in the anti-AD effects of DN200434. Further studies are warranted to determine whether ERRγ directly modulates the AKT/MAPK/NF-κB signaling pathways in keratinocytes or whether these effects are mediated via indirect mechanisms. Clarifying this will be critical to fully elucidate the mode of action of DN200434 in AD. Furthermore, we are actively pursuing mechanistic studies using immune cell profiling, cytokine network analyses, and transcriptomic approaches to better understand how ERRγ inhibition may influence specific immune pathways in both AD and other inflammatory diseases.

## 4. Materials and Methods

Information on materials and methods is provided in the Appendix A.

### 4.1. Animals

Detailed information about animals was described in Appendix A.

### 4.2. Cells

Detailed information about cells was described in Appendix A.

### 4.3. Cell Viability Assay

The viability of HaCaT cells treated with DN200434 was evaluated using a Cell Counting Kit-8 (Dojindo Molecular Technologies, Rockville, MD, USA), as described previously [16].

### 4.4. qRT-PCR and IgE Assay

Total RNA was extracted from HaCaT cells and AD dorsal skin tissues, and ERRγ, IL-6, and TNF-α mRNA expression was evaluated as described previously [16].

Serum IgE was determined by mouse IgE ELISA kits (Biolegend, San Diego, CA, USA). The procedures were carried out as per the manufacturer’s guidelines.

### 4.5. Western Blot

Total protein was extracted from HaCaT cells and AD dorsal skin tissues, and ERRγ, pAKT, pERK, p-p38, p-NFκB (Cell signaling, Danvers, MA, USA; dilution 1:1000; dilution 1:1000), GAPDH, and β-actin were evaluated as described previously [16].

### 4.6. In Vivo Study

Detailed information about in vivo study was described in Appendix A. The severity of dermatitis was assessed as described in the report by Hanifin et al. [24].

### 4.7. Histopathological Evaluation

Detailed information about histopathological analysis was described in Appendix A.

### 4.8. Statistical Analysis

Statistical analyses were conducted using PRISM software (version 10), and significance was assessed via an unpaired Student’s *t*-test, with *p*-values less than 0.05 considered statistically significant.

## Figures and Tables

**Figure 1 ijms-26-06959-f001:**
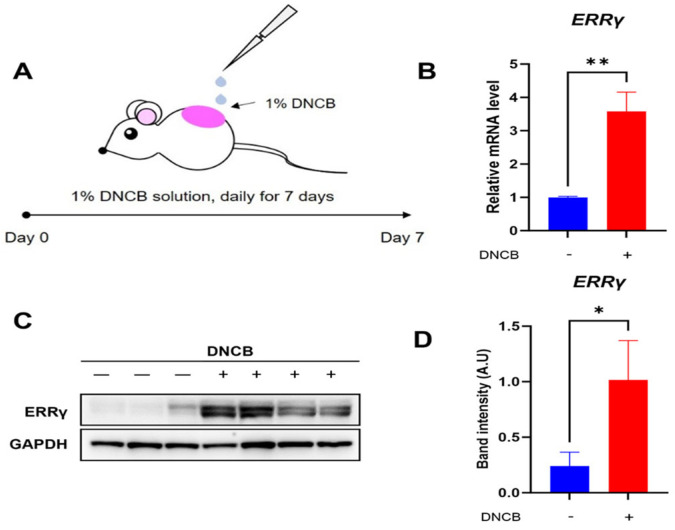
Expression of ERRγ in DNCB-induced AD lesions. (**A**) Schematic for induction of DNCB-induced AD. Expression levels of (**B**) ERRγ mRNA and (**C**,**D**) ERRγ protein in skin treated with either vehicle or DNCB. Either vehicle or 1% DNCB solution are topically treated on dorsal lesions of mice daily for seven days, followed by removal of inflamed lesions. ERRγ protein levels were normalized to GAPDH as a loading control. Data are presented as the mean ± SD; *, *p* < 0.05, **, *p* < 0.01, compared with the control.

**Figure 2 ijms-26-06959-f002:**
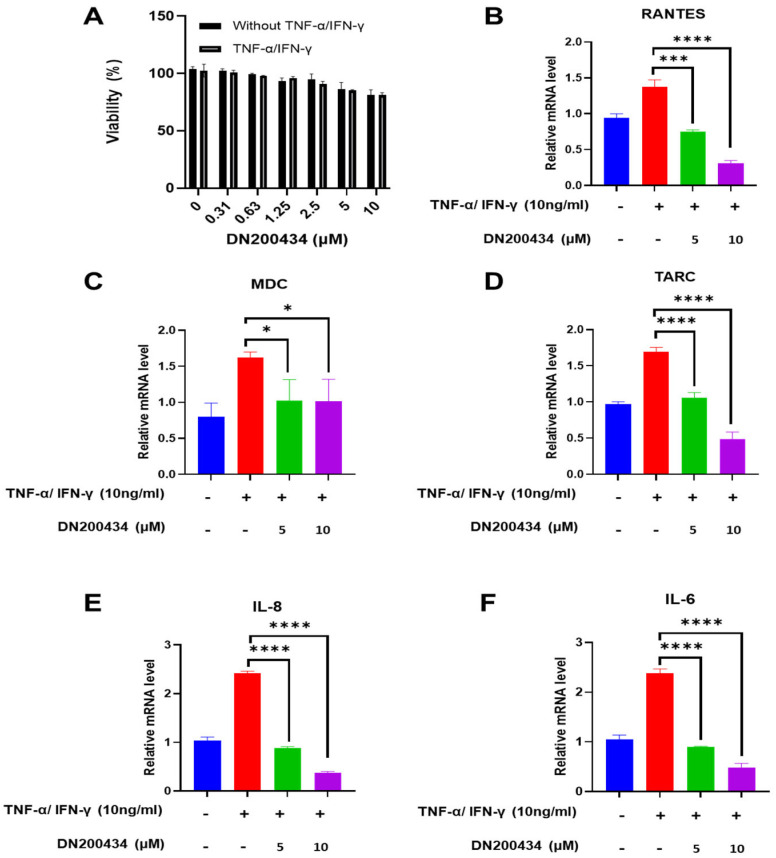
Regulation of chemokine and cytokine mRNA expression by DN200434 in TNF-α/IFN-γ-treated HaCaT cells. (**A**) Cell viability of HaCaT cells treated with or without TNF-α and IFN-γ at varying concentrations of DN200434. (**B**–**F**) DN200434-mediated down-regulation of chemokines and cytokines mRNA expression such as RANTES, MDC, TARC, IL-8, and IL-6 in TNF-α/IFN-γ-treated HaCaT cells. Data are presented as the mean ± SD; *, *p* < 0.05, ***, *p* < 0.0005, ****, *p* < 0.0001 compared with the control.

**Figure 3 ijms-26-06959-f003:**
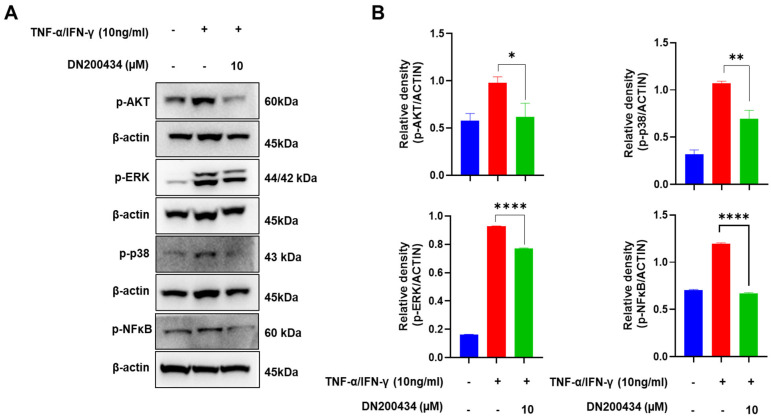
Inhibitory effect of DN200434 on AKT/MAPK/NF-κB signaling pathway in TNF-α/IFN-γ-treated HaCaT cells. (**A**) Western blot showing changes in the level of protein expression in TNF-α/IFN-γ-treated HaCaT cells by DN200434. (**B**) Graphs depicting results of quantitative analysis based on protein levels by a calibrated densitometer. Data are expressed in arbitrary units. *, *p* < 0.05, **, *p* < 0.01, ****, *p* < 0.0001 compared with the control.

**Figure 4 ijms-26-06959-f004:**
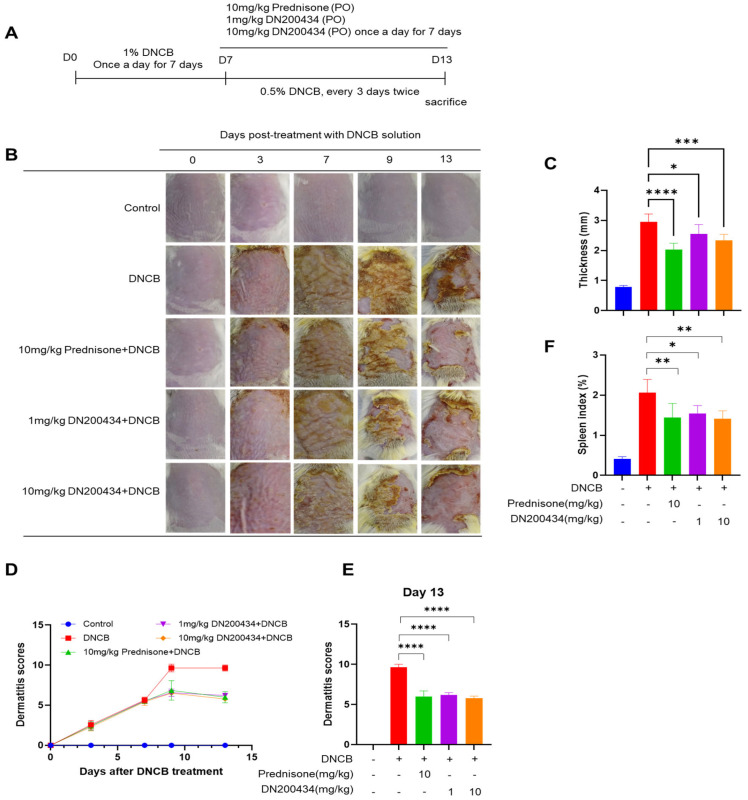
Effect of DN200434 on clinical symptoms of DNCB-induced AD. (**A**) Schematic for evaluation of anti-AD effects of DN200434 in DNCB-induced AD mice. (**B**) Representative image of mice with DNCB-induced AD. (**C**) Measurement of dorsal thickness on Day 13. (**D**) Dermatitis scores during in vivo experiments. (**E**) Bar graph showing the dermatitis score at day 13 after DNCB exposure. (**F**) Bar graph showing the spleen index day 13 after DNCB exposure. Data are presented as the mean ± SD; *, *p* < 0.05, **, *p* < 0.01, ***, *p* < 0.005, ****, *p* < 0.0001 compared with the control.

**Figure 5 ijms-26-06959-f005:**
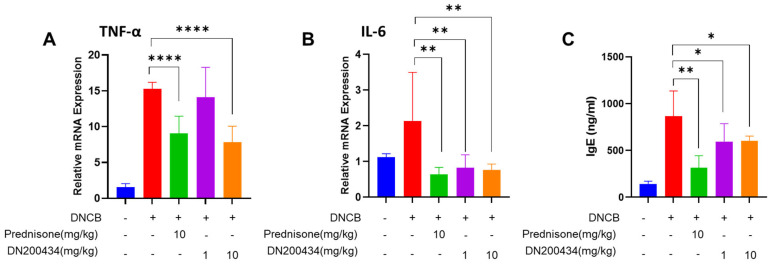
(**A**,**B**) mRNA levels of TNF-α and IL-6 in DNCB-induced AD skin lesions of mice treated with or without DN200434. (**C**) Serum level of IgE. Data are presented as the mean ± SD. Data are presented as the mean ± SD; *, *p* < 0.05, **, *p* < 0.01, ****, *p* < 0.0001 compared with the control.

**Figure 6 ijms-26-06959-f006:**
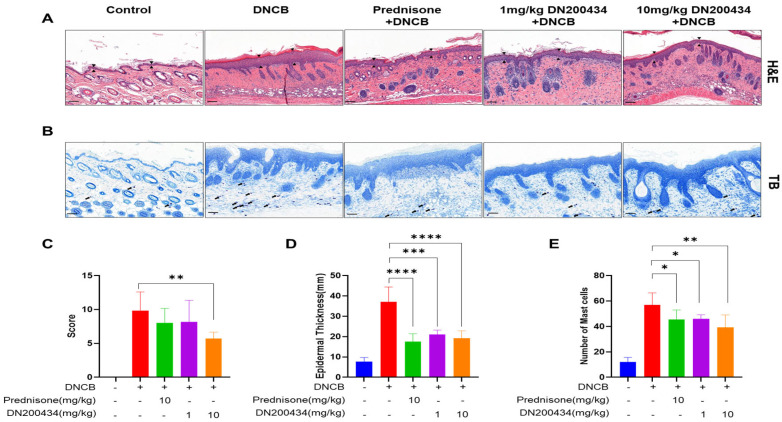
Histopathological findings in DNCB-induced AD lesions following DN200434 treatment. (**A**,**B**) Histologic findings in skin tissue section stained by H&E and toluidine blue staining (upper and lower panels, bar: 100 µm). (**C**) Histological scores. (**D**) Measurement of epidermal thickness using Image J (Version 1.54p, National Institutes of Health, Bethesda, MD, USA; available at: https://imagej.nih.gov/ij/, accessed on 17 February 2025). (**E**) A bar graph showing the mast cell numbers in Figure 6B. Data are presented as the mean ± SD; *, *p* < 0.05, **, *p* < 0.01, ***, *p* < 0.005, ****, *p* < 0.0001 compared with the control.

## Data Availability

The data that support the findings of this study are available from the corresponding author upon reasonable request. Some data may not be made available because of privacy or ethical restrictions.

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
