# Peer review of "Exploring the Therapeutic Potential of Estrogen-Related Receptor γ Inverse Agonists in Atopic Dermatitis-like Lesions"

_ijms, 2025, doi:10.3390/ijms26146959_

Round 1

Reviewer 1 Report

Comments and Suggestions for Authors

1 Mechanistic Insight

  • The manuscript claims that DN200434 exerts its anti-AD effect via inhibition of the AKT/MAPK/NF-κB signaling pathway. However, it remains unclear whether this is a direct or indirect effect of ERRγ modulation. Evidence linking ERRγ to these pathways in keratinocytes or AD models should be more thoroughly discussed or experimentally validated. The current discussion is speculative.

2 Validation of Target Engagement

  • While the authors measured ERRγ mRNA and protein expression levels in AD lesions (Fig. 1), there is no direct evidence that DN200434 modulates ERRγ activity in the in vivo model. For a compound proposed as an inverse agonist, a luciferase reporter assay or ChIP-qPCR of ERRγ target genes would strengthen the claim of target engagement.

3 Use of Positive Controls

  • Prednisone is included as a positive control in the in vivo model but is not clearly represented or discussed in all relevant figures (e.g., Fig. 5, 6). For robust comparative interpretation, the anti-inflammatory effects of DN200434 should be discussed alongside those of prednisone across all assays.

4 Figure Clarity and Densitometry

  • Western blot figures (e.g., Fig. 1C, Fig. 3A) are difficult to interpret due to image quality and missing molecular weight markers. Also, quantitative densitometry for Fig. 1C is lacking, whereas it is provided for Fig. 3. Please provide quantification of ERRγ protein levels using band intensity normalized to loading control (e.g., GAPDH) to validate overexpression in DNCB-lesioned skin.

  • From the original figures (in "ijms-3746751-original-images.pdf"), lane labeling is unclear. For example, Fig. 1C shows duplicated β-actin blots and misalignment in lane assignment. These should be corrected or clarified.

5 Histological Assessment Criteria

  • The histological score system and mast cell quantification methods (Fig. 6) lack detail. Please specify whether a blinded assessment was used, and how many fields per section were analyzed. Interobserver variability should also be addressed.

Author Response

Comments and Suggestions for Authors

1. Mechanistic Insight

(Question)

The manuscript claims that DN200434 exerts its anti-AD effect via inhibition of the AKT/MAPK/NF-κB signaling pathway. However, it remains unclear whether this is a direct or indirect effect of ERRγ modulation. Evidence linking ERRγ to these pathways in keratinocytes or AD models should be more thoroughly discussed or experimentally validated. The current discussion is speculative.

(Answer)

We appreciate the reviewer’s insightful comments. While our current study demonstrates that DN200434, a selective inverse agonist of ERRγ, reduces the activation of the AKT/MAPK/NF-κB signaling pathways in an AD mouse model, we acknowledge that the mechanistic link between ERRγ modulation and these signaling pathways remains to be clearly defined.

The original purpose of this study was to evaluate the feasibility of ERRγ as a therapeutic target for AD, as well as the therapeutic efficacy of its inverse agonist, DN200434. We are currently conducting ongoing investigations to determine whether ERRγ modulates the AKT/MAPK/NF-κB signaling pathway directly or indirectly in the context of AD, using various molecular and cellular analysis techniques.

Although we do not yet have experimental data directly linking ERRγ to these pathways in keratinocytes or in AD models, we have revised the Discussion section to explicitly highlight this limitation and emphasize the need for further mechanistic studies. Ongoing experiments using molecular and cellular approaches are currently underway to address this question. We appreciate the reviewer’s understanding and believe that these future investigations will further elucidate the mode of action underlying the anti-AD effects of DN200434.

2. Validation of Target Engagement

(Question)

While the authors measured ERRγ mRNA and protein expression levels in AD lesions (Fig. 1), there is no direct evidence that DN200434 modulates ERRγ activity in the in vivo model. For a compound proposed as an inverse agonist, a luciferase reporter assay or ChIP-qPCR of ERRγ target genes would strengthen the claim of target engagement.

(Answer)

Thank you for your valuable comments. As the reviewer correctly pointed out, while our current manuscript demonstrates that DN200434 reduces ERRγ mRNA and protein expression in AD lesions (Fig. 1), we did not include direct in vivo evidence of ERRγ target engagement such as a luciferase reporter assay or ChIP-qPCR.

However, we have previously reported that DN200434 functions as an inverse agonist of ERRγ using a luciferase reporter assay driven by an ERRγ-responsive element in vitro (1). These prior results support the compound’s mechanism of action as an ERRγ modulator at the transcriptional level.

For the reviewer’s reference, the relevant results are provided below.

(Reference)

Singh TD, Song J, Kim J, Chin J, Ji HD, Lee J-E, et al. A novel orally active inverse agonist of Estrogen-related Receptor Gamma (ERRγ), DN200434, a booster of NIS in anaplastic thyroid cancer. Clinical Cancer Research 2019; 25: 5069-81.

Materials and methods for reviewer only

Plasmids

pFR-luc (Stratagene) was used as a Gal4-driven luciferase reporter. Expression vectors for wild-type ERRγ ligand binding domain (LBD) were constructed in pCMX-Gal4DBD (DNA binding domain).

Cell culture, transient transfection, and luciferase assay

HEK 293T cells were cultured in Dulbecco’s Modified Eagle’s Medium (DMEM) supplemented with 10% fetal bovine serum (FBS, Gibco). Cells were transiently transfected with Lipofectamine 2000 (Invitrogen) according to the manufacturer’s instruction. Luciferase activity was measured after treatment with vehicle, GSK5182 (1µM), and DN200434 for 18h and normalized to b-galactosidase activity.

Results for reviewer only

Luciferase reporter assay with ERRγ mutants

To assess whether DN200434 modulates ERRγ activity, we performed an in vitro luciferase reporter assay using a wild-type ERRγ-specific promoter system. Cells transfected with the ERRγ-responsive promoter exhibited strong luciferase activity. However, treatment with either DN200434 or GSK5182 (an ERRγ inverse agonist) significantly reduced the elevated ERRγ activity.

Figure legend

Figure for Reviewer only. Inhibitory effect of GSK5182 and DN200434 on wild-type ERRγ. HEK293T cells were transfected with Gal4-luc reporter and expression vector for Gal4-ERRγ. Cells were treated with GSK5182 (1μM) or DN200434 (1μM) for 18h and then luciferase reporter assay was performed. Error bars represent mean. Compared with ERRγ-wt, respective mutant forms ***, p< 0.0001.

3. Use of Positive Controls

(Question)

Prednisone is included as a positive control in the in vivo model but is not clearly represented or discussed in all relevant figures (e.g., Fig. 5, 6). For robust comparative interpretation, the anti-inflammatory effects of DN200434 should be discussed alongside those of prednisone across all assays.

(Answer)

Thank you for your constructive comment. As the reviewer suggested, we have revised the Results and Discussion sections to clearly describe and compare the therapeutic effects of prednisone, used as a positive control, with those of DN200434 in the in vivo model. We ensured that the anti-inflammatory effects of both compounds are now consistently addressed across all relevant assays, including those shown in Figures 5 and 6.

The revised sections have been highlighted in red for ease of review. Please refer to the updated manuscript for details.

4. Figure Clarity and Densitometry

(Question)

Western blot figures (e.g., Fig. 1C, Fig. 3A) are difficult to interpret due to image quality and missing molecular weight markers. Also, quantitative densitometry for Fig. 1C is lacking, whereas it is provided for Fig. 3. Please provide quantification of ERRγ protein levels using band intensity normalized to loading control (e.g., GAPDH) to validate overexpression in DNCB-lesioned skin.

From the original figures (in "ijms-3746751-original-images.pdf"), lane labeling is unclear. For example, Fig. 1C shows duplicated β-actin blots and misalignment in lane assignment. These should be corrected or clarified.

(Answer)

Thank you for your helpful comments. As suggested by the reviewer, we have improved the image quality of the Western blots in Figures 1C and 3A and added molecular weight markers for each protein. In addition, we have now included the quantitative densitometric analysis of ERRγ protein levels in Figure 1C, normalized to GAPDH as the loading control, to validate its overexpression in DNCB-lesioned skin.

Furthermore, in the figures containing the original blots (provided in the revised supplementary file), each lane has been clearly labeled. The β-actin blots have also been corrected and consistently shown as loading controls for the respective target proteins. We have addressed the previously unclear or misaligned bands and ensured that the presentation is accurate and interpretable.

5. Histological Assessment Criteria

(Question)

The histological score system and mast cell quantification methods (Fig. 6) lack detail. Please specify whether a blinded assessment was used, and how many fields per section were analyzed. Interobserver variability should also be addressed.

(Answer)

Thank you for your valuable comment. The detailed methodology for histological scoring and mast cell quantification is provided in the Supplementary Information. As described therein, all histological assessments were performed in a blinded manner by three independent and experienced observers. For each tissue section, three randomly selected fields were analyzed per slide.

Although formal statistical analysis of interobserver variability was not performed, any discrepancies were discussed and resolved by consensus among the three observers to ensure consistency and objectivity in the evaluation.

Reviewer 2 Report

Comments and Suggestions for Authors

1. Target Identification and Therapeutic Implications: The study identifies ERRγ as a promising therapeutic target for atopic dermatitis (AD), significantly contributing to the understanding of the receptor’s role in inflammation and skin disorders. The increased expression of ERRγ observed in DNCB-induced AD lesions suggests that targeting this receptor with an inverse agonist like DN200434 may ameliorate inflammatory responses, paving the way for the development of new treatment strategies for AD.

2. Mechanistic Insights into DN200434's Action: The work highlights that DN200434 exerts its anti-inflammatory effects through the modulation of critical signaling pathways, specifically the AKT/MAPK/NFκB pathways. This finding provides essential mechanistic insights into how ERRγ modulation can alter cellular responses in inflammatory skin conditions, suggesting a potential model for exploring therapeutic interventions.

3. Preclinical Model Validation: The effective use of DNCB-induced AD mice as a preclinical model is commendable, as this model closely mimics human AD pathology. The observed alleviation of AD symptoms and histopathological improvements in DN200434-treated mice reinforces the translational potential of this approach for future clinical applications.

4. Cytokine Profiles as Biomarkers: The significant reduction in proinflammatory cytokines such as IL-6 and TNFα, alongside decreased serum IgE levels in DN200434-treated mice, underscores the potential use of these cytokines as biomarkers for monitoring treatment efficacy in AD. Future studies could focus on correlating these biomarker changes with clinical outcomes to refine therapeutic strategies further.

5. Evaluation of Histopathological Changes: The comprehensive histopathological analysis, including assessments for epidermal hyperplasia and mast cell infiltration, provides critical validation of the histological changes associated with AD. The data suggest that DN200434 notably reduces these features, highlighting its potential as a disease-modifying agent. However, further studies are necessary to elucidate the long-term effects of DN200434 on skin architecture and function.

6. Clinical Translation and Future Directions: While the preclinical data are promising, further investigation into the pharmacokinetics, safety, and long-term effects of DN200434 in human populations is essential. Studies should also examine potential drug interactions and the precise mechanisms by which ERRγ modulation might influence various immune pathways in AD and other inflammatory conditions. These findings could ultimately lead to personalized treatment approaches for patients with AD.

Author Response

Comments and Suggestions for Authors

1. Target Identification and Therapeutic Implications: The study identifies ERRγ as a promising therapeutic target for atopic dermatitis (AD), significantly contributing to the understanding of the receptor’s role in inflammation and skin disorders. The increased expression of ERRγ observed in DNCB-induced AD lesions suggests that targeting this receptor with an inverse agonist like DN200434 may ameliorate inflammatory responses, paving the way for the development of new treatment strategies for AD.

(Answer)

Thank you for your insightful summary and interpretation.

In our study, we indeed observed a significant upregulation of ERRγ in DNCB-induced atopic dermatitis (AD) lesions, particularly in the epidermal compartment. This upregulation suggested a potential pathogenic role of ERRγ in AD, prompting us to evaluate the therapeutic potential of DN200434, a selective inverse agonist of ERRγ.

Our in vivo data demonstrated that DN200434 treatment effectively reduced AD-like symptoms, including epidermal hyperplasia, mast cell infiltration, serum IgE levels, and proinflammatory cytokine expression. These findings support the hypothesis that ERRγ plays a proinflammatory role in AD pathogenesis, and that pharmacologic inhibition of ERRγ may provide therapeutic benefits.

To our knowledge, this is one of the first studies to propose ERRγ as a viable therapeutic target in AD, offering new insights into the molecular mechanisms of skin inflammation and expanding the scope of nuclear receptor-based drug discovery in dermatology.

2. Mechanistic Insights into DN200434's Action: The work highlights that DN200434 exerts its anti-inflammatory effects through the modulation of critical signaling pathways, specifically the AKT/MAPK/NFκB pathways. This finding provides essential mechanistic insights into how ERRγ modulation can alter cellular responses in inflammatory skin conditions, suggesting a potential model for exploring therapeutic interventions.

(Answer)

Thank you for pointing out the importance of the mechanistic insights provided by our study.

As described in our results, DN200434 treatment led to a marked reduction in the phosphorylation levels of AKT, ERK, and p65 (NF-κB) in skin tissues from the DNCB-induced AD model. These pathways are known to be critically involved in the regulation of inflammatory responses and immune activation in atopic dermatitis.

While our study demonstrates that pharmacological inhibition of ERRγ by DN200434 attenuates these proinflammatory signaling cascades, we acknowledge that the direct mechanistic link between ERRγ modulation and AKT/MAPK/NF-κB inhibition remains to be fully elucidated.

3. Preclinical Model Validation: The effective use of DNCB-induced AD mice as a preclinical model is commendable, as this model closely mimics human AD pathology. The observed alleviation of AD symptoms and histopathological improvements in DN200434-treated mice reinforces the translational potential of this approach for future clinical applications.

(Answer)

We appreciate the acknowledgment of our use of the DNCB-induced AD mouse model, which is widely accepted for its ability to recapitulate key histopathological and immunological features of human atopic dermatitis, including epidermal hyperplasia, elevated serum IgE, and mast cell infiltration.

These findings validate the pharmacological efficacy of DN200434 in a well-established preclinical setting and underscore the translational relevance of ERRγ-targeted therapy in AD. We believe this model provides a robust platform for further mechanistic and therapeutic investigations toward clinical development.

4. Cytokine Profiles as Biomarkers: The significant reduction in proinflammatory cytokines such as IL-6 and TNFα, alongside decreased serum IgE levels in DN200434-treated mice, underscores the potential use of these cytokines as biomarkers for monitoring treatment efficacy in AD. Future studies could focus on correlating these biomarker changes with clinical outcomes to refine therapeutic strategies further.

(Answer)

We appreciate the emphasis on the biomarker potential of proinflammatory cytokines and serum IgE in evaluating treatment efficacy.

In future studies, we plan to explore the longitudinal relationship between these biomarker changes and clinical indices such as skin lesion scores and pruritus behavior, which may help refine patient stratification and optimize therapeutic outcomes.

5. Evaluation of Histopathological Changes: The comprehensive histopathological analysis, including assessments for epidermal hyperplasia and mast cell infiltration, provides critical validation of the histological changes associated with AD. The data suggest that DN200434 notably reduces these features, highlighting its potential as a disease-modifying agent. However, further studies are necessary to elucidate the long-term effects of DN200434 on skin architecture and function.

(Answer)

We agree that the long-term impact of DN200434 on skin architecture and barrier function remains an important area for future investigation. Although our current study focused on short-term therapeutic outcomes, we are planning extended treatment studies to assess potential effects on epidermal regeneration, skin hydration, and barrier integrity (e.g., filaggrin or loricrin expression) over time.

Such data will be crucial for determining the suitability of DN200434 for chronic administration and for fully characterizing its disease-modifying capacity.

6. Clinical Translation and Future Directions: While the preclinical data are promising, further investigation into the pharmacokinetics, safety, and long-term effects of DN200434 in human populations is essential. Studies should also examine potential drug interactions and the precise mechanisms by which ERRγ modulation might influence various immune pathways in AD and other inflammatory conditions. These findings could ultimately lead to personalized treatment approaches for patients with AD.

(Answer)

We sincerely appreciate the reviewer’s insightful comments. We fully agree that further investigations are necessary to evaluate the pharmacokinetic profile, safety, and long-term efficacy of DN200434 in human populations. While our current study focused on the preclinical efficacy of DN200434 in a DNCB-induced AD mouse model, we are currently planning additional studies to characterize the pharmacokinetics and systemic safety profile of DN200434 in both rodent and non-rodent species.

In addition, we recognize the importance of elucidating the precise immunological mechanisms underlying ERRγ modulation in AD. We are actively pursuing mechanistic studies using immune cell profiling, cytokine network analyses, and transcriptomic approaches to better understand how ERRγ inhibition may influence specific immune pathways in both AD and other inflammatory diseases. These efforts will help define the broader therapeutic potential of DN200434 and its application in precision medicine for inflammatory skin disorders. We have added a brief discussion of these future directions in the revised manuscript.

Round 2

Reviewer 1 Report

Comments and Suggestions for Authors

The authors have adequately addressed all raised concerns and provided thoughtful, well-organized responses supported by additional clarifications and data where appropriate.

  1. Regarding the mechanistic insight, the authors have clearly acknowledged the current limitation and outlined ongoing studies to clarify whether the inhibition of the AKT/MAPK/NF-κB pathway by DN200434 is a direct or indirect consequence of ERRγ modulation. The revision of the Discussion section to reflect this limitation is appropriate.

  2. In terms of target engagement, the authors have provided prior in vitro evidence of DN200434 acting as an inverse agonist of ERRγ via luciferase reporter assays, which supports the rationale for its use. While in vivo validation remains an area for future work, the current clarification and citation of previous work are acceptable for this manuscript’s scope.

  3. The comparison with prednisone as a positive control has been revised and now appears consistently throughout the Results and Discussion sections, allowing better comparative interpretation of the anti-inflammatory effects of DN200434.

  4. The issues related to Western blot clarity and quantification have been addressed by improving figure quality, adding molecular weight markers, correcting labeling, and providing densitometric analyses normalized to loading controls.

  5. For the histological assessments, the authors clarified that blinded scoring was conducted by three independent observers across three fields per slide. While interobserver variability was not statistically analyzed, the use of consensus review sufficiently addresses reproducibility concerns for this type of qualitative assessment.

Overall, the authors have responded thoroughly and appropriately. I recommend acceptance of the manuscript with minor revisions incorporated as per the responses.